# Linkage between Psychological Factors and Response to Immune Checkpoint Inhibitor Therapy: A Preliminary Study

**DOI:** 10.3390/cells12202471

**Published:** 2023-10-17

**Authors:** Miri Cohen, Yosi Shamay, Johanna Czamanski-Cohen, Katerina Shulman, Shoshana Keren Rosenberg, Mahmoud Abu-Amna, Ilit Turgeman, Ludmila Merkin Livshits, Revital Birenboim, Monica Dines, Gil Bar-Sela

**Affiliations:** 1School of Social Work, University of Haifa, Haifa 3498838, Israel; revitalbirenboim@gmail.com; 2Biomedical Engineering, Technion—Israel Institute of Technology, Haifa 3109601, Israel; yshamay@bm.technion.ac.il; 3Emili Sagol Creative Arts Therapies Research Center, School of Creative Arts Therapies, University of Haifa, Haifa 3498838, Israel; joczamanski@gmail.com (J.C.-C.); mdines@univ.haifa.ac.il (M.D.); 4Carmel Medical Center, Haifa 3436212, Israel; katerinashulman@gmail.com (K.S.); ludmilama@clalit.org.il (L.M.L.); 5Bruce Rappaport Faculty of Medicine, Technion—Israel Institute of Technology, Haifa 3109601, Israel; 6Lin Medical Center, Haifa 6688304, Israel; shoshanakr@clalit.org.il; 7Cancer Center, Emek Medical Center, Afula 1834111, Israel; mahmud_ab@clalit.org.il (M.A.-A.); ilittur@clalit.org.il (I.T.)

**Keywords:** immune checkpoint inhibitor therapy, cancer, health-related quality of life, emotional distress, cytokines, PD-1, CTLA-4

## Abstract

Substantial evidence has accumulated showing that psychological distress affects immune regulation, the response to cancer treatment, and survival. The effect of psychological parameters on the effectiveness of immune checkpoint inhibitor (ICI) treatment has not yet been studied. This preliminary study aimed to (a) examine the associations between psychological factors and responses to ICI treatment and (b) assess the associations between psychological factors and blood measures of sPD-1, sCTLA-4, and cytokines that may alter the effect of ICI treatment. The participants were 62 individuals with advanced cancer, aged 18 years or older, who were candidates for ICI treatment as a new line of treatment. The participants answered questionnaires and provided blood samples and medical data prior to the start of ICI treatment and 3 months after. Perceived health status was positively associated with better responses to ICI treatment. In the subsample of participants with biomarkers, worse health-related quality of life was associated with higher IL-6 and sCTLA-4; emotional distress and sleep difficulties were associated with higher sCTLA-4; and better perceived health was associated with lower IL-6 and TNFα. sPD-1 was not associated with psychological measures. This preliminary study found for the first time that some psychological measures could be linked to responses to cancer treatment, possibly via pro-inflammatory cytokines and sCTLA-4.

## 1. Introduction

Immune checkpoint inhibitor (ICI) therapies show remarkable benefits to patients with various cancers [1]. ICI drugs act by blocking inhibitory receptors, cytotoxic T lymphocyte antigen-4 (CTLA-4), and programmed death-1 (PD-1) on T lymphocytes and their interaction with their corresponding ligands (CD80/86 and PD-L1/-L2, respectively) on tumor cells. This blockage may lead to a reactivation of T cells in the tumor microenvironment [1]. The main ICI drugs include ipilimumab and tremelimumab for CTLA-4 inhibition; nivolumab, pembrolizumab, and cemiplimab for PD-1 inhibition; and atezolizumab, avelumab, and durvalumab for PD-L1 inhibition [1]. Recent meta-analyses have shown that ICI treatments, especially a combination of these drugs, can have a significant effect on response rate, free survival, and overall survival for various advanced cancer types (e.g., melanoma, non-small-cell carcinoma, renal cell carcinoma) [2,3].

The extent of responses to ICI is related to multiple factors, including tumor and tumor-environment factors (PD-L1 expression in tumor cells, cytokines, tumor mutational burden, personal and health factors, performance status at the start of ICI, body mass index) [4]. However, inconsistent results have also been reported regarding the associations of these factors with responses to ICI therapies [3,4].

Health-related quality of life (HRQoL) refers to cancer survivors’ perceptions of the effects of cancer and its treatments on the physical, psychological, and social domains of life [5]. Studies on the impact of ICI therapy and HRQoL have generated heterogenous findings [5,6]. Recent meta-analyses reported a substantial negative impact of ICI therapies, especially from immune-related adverse events caused by the non-specific activation of the immune system [6,7]. Nevertheless, in a meta-analysis with a total of 8341 patients from 17 randomized trials, patients receiving ICI treatment reported higher HRQoL and longer times to clinical deterioration on several patient-reported outcome scales compared with patients receiving chemotherapy, in different types of solid tumors [6].

Recent studies have demonstrated that PD-1 and CTLA-4 expressed in T cells in the tumor environment also have soluble forms (sPD-1 and sCTLA-4) in peripheral blood [8,9,10,11]. These soluble proteins can be generated from an alternative splicing of their mRNA or a proteolytic cleavage of membrane-bound proteins and were found to be biologically active [9,11].

Growing data suggest that high baseline levels of sPD-L1 in cancer patients are correlated with worse responses to ICI treatment and worse survival for multiple cancer types [8,12]. In addition, sPD-1 levels were found to be associated with systemic inflammation markers, such as CRP [9]. However, other studies have shown stable or increased sPD-1 levels post-cancer therapy [9]. For example, sPD-1 has been shown to block the PD-1/PD-L immunosuppressive pathway by binding to PD-L1 and PD-L2 [11].

The role of sCTLA-4 in response to ICI treatment has been much less studied [13,14], although its levels are increased in many cancer types, such as breast, lung, esophageal, and other cancers [15], thus supporting its possible role in T-cell-mediated immune regulation [13,14,15]. Although some studies have suggested that higher sCTLA-4 levels could be related to worse outcomes in ICI treatment [13,14], a recent study suggested that higher baseline sCTLA-4 levels were associated with better prognosis in metastatic melanoma patients [15]. In addition, this study showed an increase in sCTLA-4 levels during ipilimumab treatment, but this was also associated with higher immune-related adverse events [15].

Cytokines are a diverse group of small-cell signaling proteins, have a prominent role in regulating immune activity, and control inflammatory and anti-inflammatory processes [16]. Cytokines can induce checkpoint expression and responses to ICI therapy [16,17,18,19]. Several cytokines in peripheral blood, especially pro-inflammatory cytokines, usually increase after ICI therapy [18]. Peripheral cytokines were found to predict the outcome of ICI treatment [10,17,19,20]. For example, higher levels of pro-inflammatory cytokines (e.g., IL-6, IL-8, TNFα) in peripheral blood before treatment were usually found to predict worse responses in various cancer types, including metastatic non-small-cell lung cancer and melanoma [10,21,22,23]. However, contradictory results were also reported [16]. Other cytokines have been less studied, with mixed results regarding their associations with responses to ICI [16,23,24]. Heterogeneity in these results may be attributed to the cancer type, the treatment method, the participants’ characteristics, and the detection method [19].

Substantial evidence has accumulated that emotional distress (e.g., depression or anxiety) or sleep disturbance affect immune regulation and functioning [25,26,27,28]. In particular, researchers have found emotional distress to be associated with increased levels of inflammatory markers (such as IL-1 and IL-6) and reduced levels of regulatory cytokines (e.g., IL-10) in the blood [25,26,27,28]. Therefore, emotional distress may play a role in impaired immune responses to tumors by changing the nature of the tumor microenvironment, increasing the levels of immunosuppressive cytokines, facilitating immune cell dysfunction, and thus enhancing the tumor evasion of the immune response [26]. Accordingly, meta-analyses have revealed that emotional distress can predict cancer recurrence and recurrence-free survival in various types of cancer [29,30]. For example, a meta-analysis with a total of 282,203 breast cancer patients from 17 studies showed that emotional distress was associated with cancer recurrence, all-cause mortality, and cancer-specific mortality [30]. However, significant heterogeneity exists in these studies because of sample characteristics and different methodologies [29].

These psychoimmune regulatory processes can have profound implications for responses to ICI treatment. To the best of our knowledge, the effect of psychological parameters on the effectiveness of ICI treatment has not yet been studied. Therefore, this preliminary study aimed to (a) examine the associations between psychological factors and responses to ICI treatment and (b) assess the associations between psychological variables and blood measures of sPD-1, sCTLA-4, and cytokines that may potentially alter the effect of ICI treatment.

## 2. Materials and Methods

This study was approved by the Institutional Review Board of Haemek Medical Center (#0066-19-EMC) and Carmel Medical Center (#0101-20-CMC).

### 2.1. Participants

Participants were 62 individuals with advanced cancer recruited from the Cancer Center at Emek Medical Center in Afula and the Clalit Health Care Oncology Unit in Haifa in Haifa, Israel. Participants provided informed consent for participation before starting ICI treatment as a new line of treatment for their disease. Inclusion criteria were age 18 years or older; measurable disease according to immune-related Response Evaluation Criteria in Solid Tumors (iRECIST) criteria (i.e., tumor size and lesions are detectable); ability to understand and respond to the questionnaire in Hebrew, Arabic, or Russian; and willingness to provide blood samples. Exclusion criteria were patients who were candidates for ICI treatment therapy in the adjuvant or maintenance setting and those with known major psychiatric or cognitive disorders (e.g., schizophrenia, dementia, and intellectual disabilities). Eligible patients were approached by the medical staff of the oncology units; study aims and procedure were explained, especially that participation was voluntary and would not affect their treatment regimen; and they signed a written informed consent form.

Patients answered questionnaires, and their peripheral blood measures and medical data were collected prior to the start of ICI treatment (T0) and 3 months after (T1). In total, 62 participants responded at T0 (of whom 34 gave blood samples), and 38 participated at T1. No significant differences emerged between those who did or did not participate at T1 in demographic, cancer, and treatment variables, except for treatment line (see Appendix A).

### 2.2. Measures

Medical details included type of cancer, line of treatment, type of ICI drug, response to treatment, and overall survival since onset of ICI treatment. Performance status prior to ICI treatment was evaluated based on the Eastern Cooperative Oncology Group Performance Status Scale (ECOG PS) [31], rated by a physician with levels of performance ranging from Grade 0 (fully active) to Grade 4 (completely disabled).

Treatment benefit was examined via body CT and PET-CT and measured using iRECIST [32]. iRECIST is a standard system used to measure responses to immunotherapy. Four categories of response to treatment were aggregated into two categories: 1 = complete response, partial response, or stable disease and 2 = progressive disease.

HRQoL was assessed using the 30-item European Organization for Research and Treatment of Cancer Quality of Life Questionnaire (EORTC QLQ-C30) [33]. The EORTC QLQ-C30 features 28 items arranged in five functional subscales evaluating physical, role, emotional, cognitive, and social functioning and symptoms [33]. Responses to these 28 items are given on a 4-point Likert scale from 1 (not at all) to 4 (very much). The EORTC QLQ-C30 has shown good reliability [33]. Internal consistency (Cronbach’s alpha) for the 28-item score in the present study was 0.96 at T0 and 0.93 at T1.

Perceived health, based on one item from the EORTC QLQ-C30, was assessed on a 7-point Likert scale (1 = very poor to 7 = excellent).

Emotional distress was assessed with the 18-item anxiety and depression subscales of the Brief Symptom Inventory (BSI) [34]. Severity of symptoms was rated on a 5-point Likert scale ranging from 0 (not at all) to 4 (extremely). Internal consistency in the present study was 0.94 at T0 and 0.95 at T1.

Subjective sleep quality was assessed with the 19-item Pittsburgh Sleep Quality Index (PSQI) [35]. The index features seven component scores (subjective sleep quality, sleep latency, sleep duration, sleep efficiency, sleep disturbance, hypnotic medication use, and daytime dysfunction), each weighted equally on a 0–3-point scale from 0 (not experienced during last month) to 3 (experienced three times a week or more). Internal consistency in the present study was 0.86 at T0 and 0.92 at T1.

To measure cytokines, sPD-1, and sCTLA-4 levels in peripheral blood, blood serum samples were encoded numerically and stored at −80 °C. TNFα, IL-2, IL-6, and IL-10 were analyzed in duplicates using the Human HS Cytokine A Premixed Mag Luminex Performance Assay (R&D SYST, Minneapolis, MN, USA), which is a multiple analyte detection system. PD-1 and CTLA4 were analyzed using the MILLIPLEX^®^ Human Immuno-Oncology Checkpoint Protein Panel 1–Immuno-Oncology Multiplex Assay (Millipore, Burlington, MA, USA).

### 2.3. Data Analysis

Descriptive statistics were calculated. Means, standard deviations (SDs), the ranges of the study variables at T0 and T1. Differences between timepoints were examined using paired-sample *t*-tests. Thereafter, bivariate correlations among the background and study variables were examined. To assess the adjusted associations between the psychological variables and response to treatment, logistic regression analysis was conducted. As for the blood measures, they were first standardized; then, multiple linear regression analyses were conducted to examine their unadjusted and adjusted associations with the psychological variables. Effect sizes for *t*-tests (Cohen’s *d*) and multiple regression (Cohen’s ƒ^2^) were examined. Results were considered statistically significant at *p* < 0.05; *p* < 0.06 was considered marginally significant.

## 3. Results

The presentation of the data is divided into three parts. In the first part, the demographic and medical characteristics and the means and SDs of the study variables are described. In the second part, the findings are presented regarding the relationships between the psychological variables and response to treatment for the entire sample at T0 (*N* = 62) and T1 (*N* = 38). In the third part, the relationships between the peripheral blood parameters and the psychological variables and response to treatment for the subsample (*N* = 34) are presented. The participants who participated at T1 or did not differ in demographic or medical details (Appendix A).

### 3.1. Sample Characteristics and Means (SDs) of the Study Variables at T0 

The demographic and medical details of the participants are presented in Table 1. The participants’ mean age was about 70, almost 70% of the sample was male, and 67% were married or in a relationship. Most participants were diagnosed with lung, melanoma, renal, breast, or skin squamous-cell carcinoma cancers. The ECOG PS values at the start of ICI treatment were mostly Grades 0 and 1, indicating good functioning status. The participants received either the CTLA4 + PD1 or PD1/PD-L1 drug; for most participants, it was their first episode of ICI treatment. Complete, partial, or stable disease was recorded for 50.0% of participants.

The means (and SDs) of the study variables at T0 and T1 and differences between the timepoints are described in Figure 1. At both timepoints, perceived health, HRQoL, and quality of sleep were medium, and symptoms of emotional distress were low. These measures were not significantly different between the timepoints. The associations between the psychological variables were positive, high, and significant (Appendix A).

### 3.2. Differences in Response to Treatment by Psychological Variables

Logistic regression analyses were conducted to predict responses to treatment based on psychological variables at T0 and T1 (Table 2). Line of treatment was significantly associated with response to treatment (*B* = −0.93, *SE* = 0.42, *OR* = 0.40, 95% CI [0.17, 0.90], *p* = 0.03). The ECOG PS was not significantly associated with response to treatment (*B* = 0.22, *SD* = 0.33, *OR* = 1.24, 95% CI [0.65, 2.37], *p* = 0.51). Given the central effect of these variables PS on response to treatment, both the line of treatment and the ECOG PS were controlled. Other demographic and medical variables were not associated with response to treatment. Table 2 shows that only perceived health at T0 was significantly associated with a better response to ICI treatment. Thus, an increase of one unit in perceived health status increased the likelihood of having a positive response to treatment by 36% (1 − *OR*).

### 3.3. Associations between Psychological Variables and Blood Measures

Part 2 was conducted with a subsample of 34 participants at T0 and 15 participants at T1. For the subsample of 15 participants who had blood measures for both timepoints, levels of sPD-1 significantly increased (Cohen’s *d* = 2.51, indicating a large effect size). Levels of sCTLA-4 significantly decreased between T0 and T1 (Cohen’s *d* = 0.98, indicating a large effect size). No differences were found in cytokine levels between the timepoints (Figure 1).

As for associations among the blood measures (T0), the sPD-1 and sCTLA-4 levels were strongly and positively correlated (*r* = 0.50, *p* < 0.001), and IL-6 was positively correlated with TNFα (*r* = 0.47, *p* < 0.001). No other significant associations were found among the blood measures (see Appendix A).

Multiple linear regression analyses were conducted to examine the associations between psychological variables and blood measures. First, the associations between the demographic and medical data and blood parameters were assessed. Only the ECOG PS and age were marginally associated with levels of IL-6 (*r* = 0.31, *p* = 0.07 and *r* = 0.30, *p* < 0.09, respectively); nevertheless, these were controlled in the regression analyses. The results are shown in Table 3. Significant unadjusted and adjusted negative associations were found between perceived health and TNFα and IL-6: better reported perceived health was linked with lower levels of these cytokines (Cohen’s ƒ^2^ = 0.05 and 0.09, respectively, indicative of small effect sizes). Unadjusted and adjusted significant positive associations were found between HRQoL and TNFα (Cohen’s ƒ^2^ = 0.09), IL-6 (Cohen’s ƒ^2^ = 0.18), and sCTLA-4 (Cohen’s ƒ^2^ = 0.09). Worse HRQoL was related to higher levels of the two pro-inflammatory cytokines and lower sCTLA-4 in blood (Cohen’s ƒ^2^ = 0.07, small effect sizes). In addition, emotional distress and sleep difficulties were significantly and negatively associated with sCTLA-4, such that higher distress and sleep difficulties were associated with lower sCTLA-4 (Cohen’s ƒ^2^ = 0.08, small effect size). The association was marginally significant for emotional distress. The other correlation coefficients showed similar directions but were statistically nonsignificant.

## 4. Discussion

Our preliminary results show that perceived health status was positively associated with better responses to ICI treatment. In the subsample of participants with biomarkers, worse HRQoL was associated with higher IL-6 and sCTLA-4; emotional distress and sleep difficulties were associated with higher sCTLA-4; and better perceived health was associated with lower IL-6 and TNFα. Therefore, we found that some psychological measures were linked to pro-inflammatory cytokines and sCTLA-4.

The mean scores of the psychological variables indicate a relatively positive psychological and HRQoL state, despite the advanced cancer stage of the participants. These measures were stable between pretreatment and 3 months later, which indicates a low level of physical and psychological symptoms during the first 3 months of ICI treatment. No previous studies have examined these psychological and HRQoL variables among individuals receiving ICI treatment.

Among the psychological factors examined in this study, perceived health was the only predictor of response to ICI treatment (controlling for line of treatment and the ECOG PS at baseline). Perceived health is a subjective view of individuals’ health that represents personal experiences with physical limitations and symptoms such as fatigue and pain, but it also has been found to reflect mental properties, such as feelings of distress, depression, emotional well-being, and positive affect [36,37,38,39]. These positive emotions were previously found to be an especially strong predictor of positive perceived health [37]. In support of the present results, perceived health was previously found to predict future morbidity and physical decline and mortality [40,41,42,43]. However, in light of the small sample and variability in cancer- and ICI treatment-related factors, these results should be considered with caution, and further studies are needed to reexamine the results.

Levels of biomarkers, especially cytokines, vary among healthy individuals, and inconsistencies have been reported in some studies. Moreover, technical and contextual differences make comparisons of cytokine immunoassays between studies difficult to interpret [44]. However, a comparison of the cytokine levels obtained in the present study with healthy individuals in a similar age range shows that their means and ranges are similar to those of healthy individuals [45]. The sPD-1 and sCTLA-4 of healthy individuals widely differed between studies [46]. Levels of sPD-1 in the present study were substantially higher than in healthy controls with a similar mean age—about 500 pg/mL. In contrast, levels of sCTLA-4 were about 50 pg/mL in healthy controls, thus similar to the levels measured at T0 in our study.

Worse HRQoL was associated with higher levels of pro-inflammatory TNFα and IL-6, and perceived health was negatively associated with IL-6, whereas the associations of emotional distress and sleep difficulties with these cytokines were in the same direction but did not achieve statistical significance. These results coincide with many studies demonstrating higher inflammatory biomarkers in relation to negative emotional symptoms, lower quality of life, and more sleep difficulties in general [25,28] and among cancer patients (especially IL-6 and TNFα) [26,47]. In contrast, better perceived health was previously found to be associated with lower IL-6, as reported regarding various positive psychological and well-being indicators [25]. Higher inflammation related to emotional distress may explain the findings of higher recurrence and mortality rates among distressed cancer patients [29,30].

Worse HRQoL, more sleep difficulties, and higher emotional distress (marginal statistical significance) were associated with lower levels of sCTLA-4. Thus, negative emotional symptoms and sleep difficulties were correlated with lower sCTLA-4 at baseline, in contrast to their connection with pro-inflammatory cytokines. Given the lack of data on the role of sCTLA-4 in response to ICI treatment, the meaning of these results should be further examined.

The finding that levels of PD-1 substantially increased and levels of sCTLA-4 decreased between baseline and 3 months of ICI treatment should be reviewed with caution given the decrease in the number of participants with blood measures at T1. A prior study assessed changes in PD-1 levels during ICI treatment and supported the finding regarding increased sPD-1 levels from baseline to T1, suggesting that the main increase occurs at the start of ICI treatment [12]. These results can be explained by the activation of PBMCs, including T lymphocytes, by PD-1 antibody therapy [12]. As for sCTLA-4, its levels decreased with treatment, in contrast to Pistillo et al.’s [15] study, which reported a dose-response increase in sCTLA-4 with number of cycles. These findings correspond to the conclusions of several studies and reviews finding that the biological function and effects of ICI treatment on soluble PD-1 and CTLA4 have yet to be elucidated [8,11,12,13,14]. The findings showing that levels of pro-inflammatory cytokines did not change between baseline and T1 correspond to contradictory previous results regarding the change and predictive role of pro-inflammatory cytokine in response to ICI treatment, e.g., [10,16,21,22,23,24].

Given the preliminary nature of these results, it may be too early to suggest clinical implications. However, the results strengthen the notion that early psychosocial interventions to prevent symptoms such as sleep difficulties and to increase positive health perceptions may improve response to treatment. Thus, intensive psychosocial care for patients receiving ICI therapy is warranted.

This was a preliminary study with several limitations. The main limitation is the small sample size, preventing us from controlling for medical and psychosocial factors. In addition, the current study was a pilot study and, as such, did not include a healthy control group. An additional limitation is the inclusion of patients with various types of cancer. Monitoring psychological factors periodically may not provide real-life information on mood, perceptions, or sleep changes between assessment periods.

## 5. Conclusions

This study reported for the first time on the effect of perceived health—a cognitive appraisal that combines perceptions of physical health and psychological well-being—on responses to ICI treatment. In addition, it is the first to report the connection between blood measures, sPD-1, sCTLA-4, and cytokines and psychological factors, and it is one of few studies reporting changes in levels of blood measures from baseline to T1 during ICI treatment. The meaning of these findings and their relevance for responses to ICI treatment and survival should be further studied. To answer the open questions raised in the present study, it is imperative to study larger samples with a longitudinal procedure and assess the effects of multiple factors related to cancer treatment and personal characteristics.

## Figures and Tables

**Figure 1 cells-12-02471-f001:**
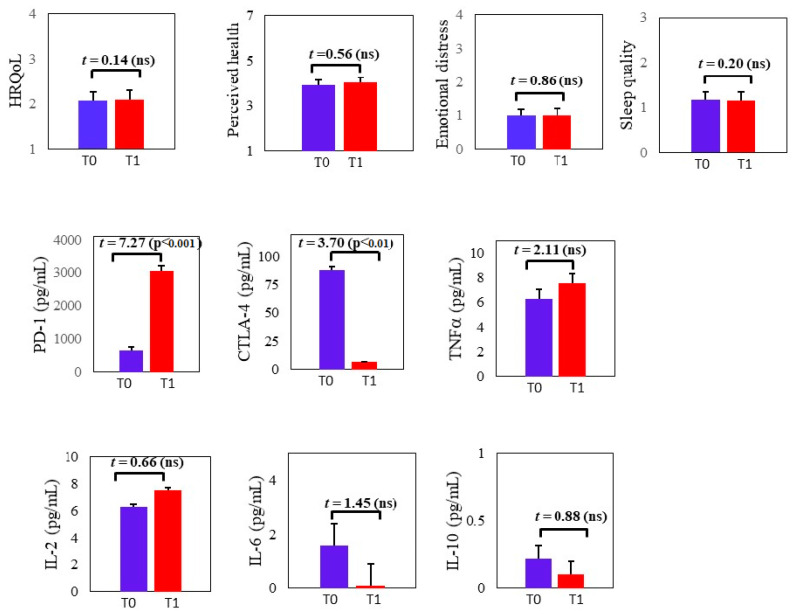
Means (SDs) of study variables and differences by time.

**Table 1 cells-12-02471-t001:** Demographic and medical details of participants.

	*M* or *n*	*SD* or %	Range
Age (years, *M*, *SD*)	69.58	13.63	37–97
Sex (*n*, %)			
Female	19	30.6	
Male	43	69.4	
Education (years, *M*, *SD*)	12.60	2.98	7–20
Marital status (*n*, %)			
Married or partnered	42	67.7	
Divorced	11	17.8	
Widowed	8	12.9	
Single	1	1.6	
Income level ^a^ (*n*, %)			
High	4	6.9	
Average	35	60.3	
Low	19	32.8	
Religion (*n*, %)			
Jewish	51	82.3	
Arab (Muslim or Christian)	9	14.5	
Other	2	3.2	
Tumor type (*n*, %)			
Lung	18	29.0	
Melanoma	14	22.6	
Renal	8	12.9	
Breast	6	9.7	
Skin small-cell carcinoma	6	9.7	
Urinary bladder	4	6.5	
Colon	3	4.8	
Gastric	2	3.2	
Sarcoma	1	1.6	
ECOG PS at T0 (*n*, %)			
0	29	46.8	
1	22	35.5	
2	11	17.7	
Type of ICI treatment (*n*, %)			
CTL4 + PD1 ^b^	23	37.1	
PD1/PDL1 ^c^	39	33.8	
Line of IT (*n*, %)			
1	40	64.5	
2	16	25.8	
3	6	9.7	
Best response to IT (*n*, %)			
Complete response	6	9.7	
Partial response	18	29.0	
Stable disease	7	11.3	
Progressive disease	31	50.0	
Overall survival time (deceased) ^d^ (months, *M*, *SD*)	217.75	236.62	20–974
Overall survival (alive) ^e^ (Months, *M*, *SD*)	573.91	274.14	141–1354

^a^ Calculated from actual responses. ^b^ Ipilimumab and nivolumab. ^c^ Pembrolizumab, nivolumab, cimiplimab, atezolizumab, durvalumab. ^d^ From start of treatment to death, in days. ^e^ From start of treatment to last assessment, in days.

**Table 2 cells-12-02471-t002:** Logistic regression of response to treatment by psychological variables (T0 and T1).

Variable	Unadjusted *OR* (95% CI)	*p*	Adjusted *OR* (95% CI)	*p*
Perceived health status (T0)	0.67 (0.47, 0.97)	**0.03**	0.65 (0.42, 0.99)	**0.04**
Perceived health status (T1)	0.85 (0.54, 1.32)	0.46	0.63 (0.30, 1.31)	0.21
Emotional distress (T0)	1.44 (0.71, 2.87)	0.31	1.58 (0.65, 3.85)	0.31
Emotional distress (T1)	1.53 (0.67, 3.52)	0.31	2.15 (0.75, 6.11)	0.15
HRQoL (T0)	1.13 (0.55, 2.33)	0.74	1.11 (0.45, 1.76)	0.60
HRQoL (T1)	0.95 (0.36, 1.48)	0.81	1.16 (0.32, 4.30)	0.83
Sleep difficulties (T0)	1.10 (0.49, 2.44)	0.82	1.11 (0.44, 2.77)	0.40
Sleep difficulties (T1)	1.38 (0.48, 3.99)	0.56	1.93 (0.52, 7.08)	0.32

Bold indicates *p* < 0.05. Note: response to treatment: 1 = complete response, partial response, or stable disease; 2 = progressive disease.

**Table 3 cells-12-02471-t003:** Unadjusted and adjusted associations between psychological variables and blood measures (T0).

Variables	TNFα		IL-2		IL-6		IL-10		PD-1		CTLA-4	
	Unadj. (*p*)	Adj. (*p*)	Unadj. (*p*)	Adj. (*p*)	Unadj. (*p*)	Adj. (*p*)	Unadj. (*p*)	Adj. (*p*)	Unadj. (*p*)	Adj. (*p*)	Unadj. (*p*)	Adj. (*p*)
Perceived health	**−0.35 (0.04)**	**−0.43 (0.03)**	−0.24 (0.23)	−0.12 (0.38)	**−0.39 (0.02)**	**−0.39 (0.04)**	−0.12 (0.31)	−0.10 (0.61)	0.16 (0.38)	0.13 (0.52)	0.17 (0.34)	0.21 (0.41)
HRQoL	**0.32 (0.06)**	**0.45 (0.02)**	0.27 (0.15)	0.16 (0.45)	**0.48 (0.04)**	**0.50 (0.00)**	0.08 (0.57)	0.06 (0.80)	−0.26 (0.13)	−0.26 (0.19)	**−0.36 (0.07)**	**−0.45 (0.04)**
Emotional distress	0.23 (0.18)	0.24 (0.20)	0.06 (0.75)	0.05 (0.80)	0.24 (0.13)	0.23 (0.12)	−0.01 (0.95)	−0.05 (0.81)	−0.17 (0.34)	−0.13 (0.50)	−0.35 (0.08)	−0.41 (0.06)
Sleep difficulties	0.07 (0.72)	0.10 (0.62)	−0.04 (0.84)	−0.11 (0.60)	0.20 (0.30)	0.11 (0.66)	0.31 (0.09)	0.37 (0.07)	−0.24 (0.19)	−0.28 (0.19)	**−0.50 (0.01)**	**−0.53 (0.02)**

Standardized coefficients (betas) are presented; standardized blood measures were used in the analysis. Bold indicates *p* < 0.05.

## Data Availability

The data are available upon request.

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
