# Peer review of "Linkage between Psychological Factors and Response to Immune Checkpoint Inhibitor Therapy: A Preliminary Study"

_cells, 2023, doi:10.3390/cells12202471_

Round 1

Reviewer 1 Report

Review - Cells

Manuscript title: Linkage between psychological factors and response to immune checkpoint inhibitors therapy: A preliminary study

ראש הטופס

The present manuscript delves into the impact of psychological parameters on the effectiveness of immune checkpoint inhibitor (ICI) treatment. The authors aim to establish a scientific basis for the connection between specific psychological measures and their influence on the response to cancer treatment, as indicated by blood measures such as sPD-1, sCTLA-4, and cytokines.

This manuscript makes a significant contribution to the scientific literature, offering a unique perspective on the role of psychological factors, particularly perceived health, in shaping responses to ICI treatment. However, I have several comments and suggestions for consideration prior to publication:

Introduction: The authors discuss various psychological factors (e.g., depression, anxiety, emotional distress, sleep disturbances) known to impact immune regulation and functioning. However, the Health-Related Quality of Life (HRQoL) was omitted in the introduction, although it play a significant role in the research model. I recommend addressing this gap. Please refer to the following source as an example: Faury S, Foucaud J (2020) Health-related quality of life in cancer patients treated with immune checkpoint inhibitors: A systematic review on reporting of methods in randomized controlled trials. PLoS ONE 15(1): e0227344. https://doi.org/10.1371/journal.pone.022734

Additionally, in lines 88-92, the authors state the study's aims. Could they also propose hypotheses?

Materials and Methods: The presentation of the study procedure appears to be somewhat brief. Further elaboration on participant recruitment is recommended, including details on who approached the participants and whether participation was voluntary. Additionally, clarification is needed regarding the inclusion criteria mentioned in lines 101-102, specifically, "measurable disease according to iRECIST criteria" and "ability to understand and adhere to the study requirements."

Concerning the study participants, the authors note in lines 107-110 that "No significant differences emerged between those who did or did not participate at T1." Could they specify which variables were examined? This information may be included in the supplementary section. Furthermore, it is mentioned that "the peripheral blood measures were collected for a subsample (N=34) only." Please clarify the source sample from which this sub-sample was derived and when these blood measures were obtained.

In the Measures section, please capitalize "treatment" benefit.

Results: I have reservations regarding the adequacy of the dataset for some of the analyses conducted. While I acknowledge that this is preliminary research, certain analyses may require more than 38 participants. Presenting effect sizes could be beneficial.

In lines 183-185, the authors state that "Table 3 shows that only perceived health at T0 was significantly associated with better response to ICI treatment (in the adjusted model and marginally significant in the non-adjusted model)." However, in Table 3, the non-adjusted model is linked to p= .03. Clarification on this discrepancy is required. Additionally, it is unclear how the authors arrived at the argument: "Thus, an increase of 1 in perceived health status increased the likelihood of having a positive response to treatment by 36%" (Lines 185-187). Could they provide the source of this 36%?

Discussion: The authors mention in line 242 the association between positive affect and perceived health as part of their explanation for why perceived health predicts response to ICI treatment. It would be helpful to provide a justification for this explanation, as positive affect was not assessed in their research. How does it contribute to their interpretation of the findings?

Finally, I am curious about the implications of this preliminary research. Could the authors suggest potential directions for their study implications?

Technical Issues: The present manuscript contains spelling errors (e.g., "perihelial blood measures" - Line 109), inconsistencies in font size (e.g., line 40, Table 3, etc.), a lack of capitalization where needed (e.g., line 117), and inconsistencies in abbreviation usage (e.g., Line 171 "T0 and Time 1"). Professional editing is recommended to enhance clarity.

In conclusion, the authors address an innovative topic regarding the connection between psychological factors and the response to immune checkpoint inhibitor therapy. I recommend accepting this manuscript pending the implementation of the suggested comments.

Author Response

Reviewer #1

The present manuscript delves into the impact of psychological parameters on the effectiveness of immune checkpoint inhibitor (ICI) treatment. The authors aim to establish a scientific basis for the connection between specific psychological measures and their influence on the response to cancer treatment, as indicated by blood measures such as sPD-1, sCTLA-4, and cytokines.

This manuscript makes a significant contribution to the scientific literature, offering a unique perspective on the role of psychological factors, particularly perceived health, in shaping responses to ICI treatment. However, I have several comments and suggestions for consideration prior to publication:

Response: We thank the reviewer for the positive feedback

Introduction: The authors discuss various psychological factors (e.g., depression, anxiety, emotional distress, sleep disturbances) known to impact immune regulation and functioning. However, the Health-Related Quality of Life (HRQoL) was omitted in the introduction, although it play a significant role in the research model. I recommend addressing this gap. Please refer to the following source as an example: Faury S, Foucaud J (2020) Health-related quality of life in cancer patients treated with immune checkpoint inhibitors: A systematic review on reporting of methods in randomized controlled trials. PLoS ONE 15(1): e0227344. https://doi.org/10.1371/journal.pone.022734

Response: We added a paragraph on HRQoL among patients receiving ICI therapy (p.4, second paragraph): Health-related quality of life (HRQoL) refers to cancer survivors’ perceptions of the effects of cancer and its treatments on physical, psychological, and social domains of life [28]. Studies on the impact of ICI therapy and HRQoL have generated heterogenous findings [29,30]. Recent meta-analyses reported a substantial impact of ICI therapies, especially due to adverse immune-related events [29,30]. Nevertheless, HRQoL was found to be higher among patients receiving ICI treatment compared to patients receiving chemotherapy [29].

Additionally, in lines 88-92, the authors state the study’s aims. Could they also propose hypotheses?

Response: This was a preliminary and exploratory study. Therefore, we believe that it is too early in the present state of knowledge regarding the research questions to outline hypotheses.

Materials and Methods: The presentation of the study procedure appears to be somewhat brief. Further elaboration on participant recruitment is recommended, including details on who approached the participants and whether participation was voluntary. Additionally, clarification is needed regarding the inclusion criteria mentioned in lines 101-102, specifically, “measurable disease according to iRECIST criteria” and “ability to understand and adhere to the study requirements.”

Response: To explain measurable iRECIST criteria, we added the following explanation in parentheses: “(i.e., tumor size and lesions are detectable).” Please note that further explanation of iRECIST is given in the Measures section (p. 6, 4th paragraph): Treatment benefit was examined by body CT and PET-CT and measured by iRECIST [32]. iRECIST is a standard system used to measure response to immunotherapy. Four categories of response to treatment were aggregated into two categories: 1 = complete response, partial response, or stable disease and 2 = progressive disease.

“Ability to understand and adhere to the study requirements” was changed to “ability to ability to understand and respond to the questionnaire in Hebrew, Arabic, or Russian and willingness to give blood samples.” (p. 5, 4th paragraph)

In addition, we added the following details on recruitment procedures (p. 6, first paragraph): “Eligible patients were approached by the medical staff of the oncology units; study aims and procedure were explained, especially that participation was voluntary and would not affect their treatment regimen; and they signed a written informed consent..”

Concerning the study participants, the authors note in lines 107-110 that “No significant differences emerged between those who did or did not participate at T1.” Could they specify which variables were examined? This information may be included in the supplementary section.

Response: We added a supplemental table (Table S1, p. ) along with a description of the variables that were compared between the two groups (p. 5, first paragraph): “No significant differences emerged between those who did or did not participate at T1 in demographic variables and cancer and treatment variables, except for treatment line (see Table S1)”.

Furthermore, it is mentioned that “the peripheral blood measures were collected for a subsample (N=34) only.” Please clarify the source sample from which this sub-sample was derived and when these blood measures were obtained.

Response: We changed the text to better clarify these points: (p. 6, first paragraph): “Patients answered questionnaires and their peripheral blood measures and medical data were collected prior to the start of ICI treatment (T0) and 3 months later (T1). Sixty-two participants responded at T0 (of whom 34 gave blood samples) and 38 participated at T1.”

In the Measures section, please capitalize “treatment” benefit.

Response: Done.

Results: I have reservations regarding the adequacy of the dataset for some of the analyses conducted. While I acknowledge that this is preliminary research, certain analyses may require more than 38 participants. Presenting effect sizes could be beneficial.

Response: We added the effect sizes for the variables with statistically significant differences between means at T0 and T1 (see p. 12, last paragraph) and for significant beta coefficients (linear regression analyses)  (p. 13, first paragraph).

“Part 2 was conducted with a subsample of 34 participants at T0 and 15 participants at T1. For the subsample of 15 participants who had blood measures for both time points, levels of sPD-1 significantly increased (Cohen’s d = 2.51, indicating a large effect size). Levels of sCTLA-4 significantly decreased between T0 and T1 (d = 0.98, indicating a large effect size). No differences were found in cytokines levels between time points (Table 2).”

“Multiple linear regression analyses were conducted to examine the associations between psychological variables and blood measures. First, the associations between the demographic and medical data and blood parameters were assessed. Only ECOG PS and age were marginally associated with levels of IL-6 (r = .31, p = .07 and r = .30, p < .09, respectively); nevertheless, these were controlled in the regression analyses. The results are shown in Table 3. Significant unadjusted and adjusted negative associations were found between perceived health and TNFa and IL-6: Better reported perceived health was linked with lower levels of these cytokines (Cohen’s ƒ2 = .05 and .09, respectively, indicative of small effect sizes). The unadjusted and adjusted significant positive associations were found between HRQoL and TNFa (Cohen’s ƒ2 = .09), IL-6 (Cohen’s ƒ2 = .18), and sCTLA-4 (Cohen’s ƒ2 = .09). Worse HRQoL was related to higher levels of the two pro-inflammatory cytokines and lower sCTLA-4 in blood (small effect sizes). In addition, emotional distress and sleep difficulties were significantly and negatively associated with sCTLA-4, such that higher distress and sleep difficulties were associated with lower sCTLA-4 (Cohen’s ƒ2 = .08, small effect size). The association was marginally significant  for emotional distress. The other correlation coefficients showed similar directions but were statistically nonsignificant.”

As for logistic regression, OR is the indicator of effect size. Because Odds ratios measure how many times bigger the odds of one outcome is for one value of an IV, compared to another value.

In lines 183-185, the authors state that “Table 3 shows that only perceived health at T0 was significantly associated with better response to ICI treatment (in the adjusted model and marginally significant in the non-adjusted model).” However, in Table 3, the non-adjusted model is linked to p= .03. Clarification on this discrepancy is required.

Response: p = .03 indicates a statistically significant result.

Additionally, it is unclear how the authors arrived at the argument: “Thus, an increase of 1 in perceived health status increased the likelihood of having a positive response to treatment by 36%” (Lines 185-187). Could they provide the source of this 36%?

Response: We noted that the percentage of change in the response is 1-OR (p. 12, line 224) The OR is lower than 1 because a better response is defined as 1 and a worse response is defined as 2. We added this explanation below Table 2 (in addition, it is explained in the Measures section (p.6, 4th paragraph: “Four categories of response to treatment were aggregated into two categories: 1 = complete response, partial response, or stable disease and 2 = progressive disease.”)

Discussion: The authors mention in line 242 the association between positive affect and perceived health as part of their explanation for why perceived health predicts response to ICI treatment. It would be helpful to provide a justification for this explanation, as positive affect was not assessed in their research. How does it contribute to their interpretation of the findings?

Response: We changed the text for more clarity (p. 15, 3rd paragraph): Perceived health is a subjective view of individuals’ health that represents personal experiences with physical limitations and symptoms such as fatigue and pain, but it also has been found to reflect mental properties, such as feelings of distress, depression, emotional well-being, and positive affect [36-39]. These positive emotions were previously found to be an especially strong predictor of positive perceived health [37].

Finally, I am curious about the implications of this preliminary research. Could the authors suggest potential directions for their study implications?

Response: We hesitated to add practical implications due to the preliminary data we gathered. However, in line with reviewer’s request, a short paragraph was added   (p.17, second paragraph): “Due to the preliminary nature of these results, it may be too early to suggest clinical implications. However, the results strengthen the notion that early psychosocial interventions to prevent symptoms such as sleep difficulties and increase positive health perceptions may improve response to treatment. Thus, intensive psychosocial care for patients receiving ICI therapy is warranted.”.

Technical Issues: The present manuscript contains spelling errors (e.g., “perihelial blood measures” - Line 109), inconsistencies in font size (e.g., line 40, Table 3, etc.), a lack of capitalization where needed (e.g., line 117), and inconsistencies in abbreviation usage (e.g., Line 171 “T0 and Time 1”). Professional editing is recommended to enhance clarity.

Response: The errors were corrected and the paper was re-edited by a professional editor.

In conclusion, the authors address an innovative topic regarding the connection between psychological factors and the response to immune checkpoint inhibitor therapy. I recommend accepting this manuscript pending the implementation of the suggested comments.

Reviewer 2 Report

In addition to the effect of a particular therapy on the target, it is also important to study its effect on the general condition of a person, including psychological. Indeed, more and more evidence is now being discovered that changes in a person’s psychological state can be directly related to changes in immune system, and vice versa. In your study “Linkage between psychological factors and response to immune checkpoint inhibitors: A preliminary study,” you attempted to study the relationship between psychological parameters and the effectiveness of immune checkpoint therapy in people with cancer. This study was performed and described at a high level. The fact that this is a preliminary study mitigates a number of limitations, such as sample size. While reading this manuscript, I had several comments, most of which were cosmetic in nature. All comments are described in more detail below.

Major concern

1) The main limitation of the study is that it lacked a healthy control group from the same demographic group at T0. The manuscript describes a variety of psychological and immune parameters and their changes after therapy. However, their relationship to normal values remains unclear. Preliminary study status can mitigate the fact that you do not have a control group. However, I would like to see in the “Discussion” section some information on the parameters you are studying in healthy donors, described in the literature.

2) Please indicate in the limitations paragraph that your study involves different types of cancer.

Minor concerns

Line 63: [10-12). Replace the parenthesis with a square bracket.

Line 117: Start a new sentence with a capital letter.

Line 130: [ 29]. Remove an extra space.

Line 156: (SDs) deviations. The SD acronym already contains the word deviation, leave SDs here, and enter the acronym in line 145, where you use standard deviations for the first time.

Line 171: Means (SDs) and. As far as I can understand, here you meant “Means and SDs”? Or is there just an extra word «and»?

Line 175; Line 196: According to Cells rules, supplementary material must be named a certain way. For tables, these are Table S1 and Table S2, respectively. Also change this in the supplementary materials themselves.

Line 195: (r=.47). Add p-value as you did in the line 194.

Line 201: In the line 153 you write that marginal significance started at values p<0.07, but on line 201 you state marginal significance for the result p<0.09.

Line 279: (e.g., 7,15,18-21]. Place the brackets in this place correctly.

Add an "Author Contributions" section. You can see its location in the text and format in any article published in Cells, or in the “Instructions for Authors” section on the journal page.

This is not a flaw per se, but rather my subjective opinion, so this comment will not influence my decision on this manuscript. However, I find the presentation of data in the form of tables only, without graphs, rather bland. You could replace one of the tables with figures to add variety to the presentation of your research findings. In my opinion, the most appropriate table for this is Table 3. However, this comment is my personal opinion and you may or may not do so. At your discretion.

Author Response

Reviewer #2

In addition to the effect of a particular therapy on the target, it is also important to study its effect on the general condition of a person, including psychological. Indeed, more and more evidence is now being discovered that changes in a person’s psychological state can be directly related to changes in immune system, and vice versa. In your study “Linkage between psychological factors and response to immune checkpoint inhibitors: A preliminary study,” you attempted to study the relationship between psychological parameters and the effectiveness of immune checkpoint therapy in people with cancer. This study was performed and described at a high level. The fact that this is a preliminary study mitigates a number of limitations, such as sample size. While reading this manuscript, I had several comments, most of which were cosmetic in nature. All comments are described in more detail below.

Response: We thank the reviewer for the positive feedback

Major concern

1) The main limitation of the study is that it lacked a healthy control group from the same demographic group at T0. The manuscript describes a variety of psychological and immune parameters and their changes after therapy. However, their relationship to normal values remains unclear. Preliminary study status can mitigate the fact that you do not have a control group. However, I would like to see in the “Discussion” section some information on the parameters you are studying in healthy donors, described in the literature.

Response: We added a section in the discussion. However, it should be noted that comparisons between studies with different populations and in different labs are difficult to interpret (p.16, first paragraph): “Levels of biomarkers, especially cytokines, vary among healthy individuals and inconsistencies have been reported among studies. Moreover, technical and contextual differences make comparisons of cytokine immunoassays between studies difficult to interpret [44]. However, a comparison of the cytokines levels obtained in the present study with healthy individuals in a similar age range shows that their means and ranges are similar to those of healthy individuals [45]. sPD-1 and sCTLA-4 of healthy individuals’ serum widely differed among studies [46]. Levels of sPD-1 in the present study were substantially higher than in healthy controls with a similar mean age—about 500 pg/ml. In contrast, levels of sCTLA-4 were about 50 pg/ml in healthy controls, thus similar to the levels measured at T0 in our study.”

We also added in the limitation section (p. 17, 3rd paragraph): “In addition, the current study was a pilot study and as such, did not include a healthy control group”.

2) Please indicate in the limitations paragraph that your study involves different types of cancer.

Response: We added the limitation that study the involved different types of cancer (p. 17, 3rd paragraph): “An additional limitation is the inclusion of patients with various types of cancer.”.

Minor concerns

Line 63: [10-12). Replace the parenthesis with a square bracket.

Line 117: Start a new sentence with a capital letter.

Line 130: [ 29]. Remove an extra space.

Line 156: (SDs) deviations. The SD acronym already contains the word deviation, leave SDs here, and enter the acronym in line 145, where you use standard deviations for the first time.

Line 171: Means (SDs) and. As far as I can understand, here you meant “Means and SDs”? Or is there just an extra word «and»?

Line 175; Line 196: According to Cells rules, supplementary material must be named a certain way. For tables, these are Table S1 and Table S2, respectively. Also change this in the supplementary materials themselves.

Line 195: (r=.47). Add p-value as you did in the line 194.

Line 201: In the line 153 you write that marginal significance started at values p<0.07, but on line 201 you state marginal significance for the result p<0.09.

Response: We deleted this line. As for the background variables, we decided to be stricter in controlling these variables although they were not significantly associated with the dependent variables.

Line 279: (e.g., 7,15,18-21]. Place the brackets in this place correctly.

Response: all correction were done

Add an “Author Contributions” section. You can see its location in the text and format in any article published in Cells, or in the “Instructions for Authors” section on the journal page.

Response: added

This is not a flaw per se, but rather my subjective opinion, so this comment will not influence my decision on this manuscript. However, I find the presentation of data in the form of tables only, without graphs, rather bland. You could replace one of the tables with figures to add variety to the presentation of your research findings. In my opinion, the most appropriate table for this is Table 3. However, this comment is my personal opinion and you may or may not do so. At your discretion.

Response: We replaced Table 2 with a figure. Table 2 seemed more appropriate to present as a figure.

Round 2

Reviewer 2 Report

Thank you for taking into account all the comments. I have no further suggestions about your manuscript. Good luck in your future work!